# The Lingering of Gradients: How to Reuse Gradients over Time

**Zeyuan Allen-Zhu**∗
Microsoft Research AI
Redmond, WA 98052
zeyuan@csail.mit.edu

**David Simchi-Levi**∗
MIT
Cambridge, MA 02139
dslevi@mit.edu

**Xinshang Wang**∗
MIT
Cambridge, MA 02139
xinshang@mit.edu

## Abstract

Classically, the time complexity of a first-order method is estimated by its number of gradient computations. In this paper, we study a more refined complexity by taking into account the "lingering" of gradients: once a gradient is computed at $x_k$, the additional time to compute gradients at $x_{k+1}, x_{k+2}, \dots$ may be reduced.

We show how this improves the running time of gradient descent and SVRG. For instance, if the "additional time" scales linearly with respect to the traveled distance, then the "convergence rate" of gradient descent can be improved from $1/T$ to $\exp(-T^{1/3})$. On the empirical side, we solve a hypothetical revenue management problem on the Yahoo! Front Page Today Module application with 4.6m users to $10^{-6}$ error (or $10^{-12}$ dual error) using 6 passes of the dataset.

## 1 Introduction

First-order methods play a fundamental role in large-scale machine learning and optimization tasks. In most scenarios, the performance of a first-order method is represented by its *convergence rate*: the relationship between $\varepsilon$ (the optimization error) versus $T$ (the number of gradient computations). This is meaningful because in most applications, the time complexities for evaluating gradients at different points are of the same magnitude. In other words, the worse-case time complexities of first-order methods are usually proportional to a fixed parameter times $T$.

In large-scale settings, however, if we have already spent time computing the (full) gradient at $x$, perhaps we can use such information to reduce the time complexity to compute full gradients at other points near $x$. We call this the "lingering" of gradients, because the gradient at $x$ may be partially reused for future consideration, but will eventually fade away once we are far from $x$.

Formally, consider the (finite-sum) stochastic convex minimization problem:

$$\min_{x \in \mathbb{R}^d} \left\{ f(x) \stackrel{\text{def}}{=} \frac{1}{n} \sum_{i=1}^n f_i(x) \right\} . \tag{1.1}$$

Then, could it be possible that *whenever* $x$ is sufficiently close to $y$, for at least a large fraction of indices $i \in [n]$, we have $\nabla f_i(x) \approx \nabla f_i(y)$? In other words, if $\nabla f_1(x), \dots, \nabla f_n(x)$ are already calculated at some point $x$, can we reuse a large fraction of them to approximate $\nabla f(y)$?

**Example 1.** In classification problems, $f_i(x)$ represents the loss value for "how well training sample $i$ is classified under predictor $x$". For any sample $i$ that has a large margin under predictor $x$, its gradient $\nabla f_i(x)$ may stay close to $\nabla f_i(y)$ whenever $x$ is close to $y$.

Formally, let $f_i(x) = \max\{0, 1 - \langle x, a_i \rangle\}$ be the hinge loss (or its smoothed variant if needed) with respect to the $i$-th sample $a_i \in \mathbb{R}^d$. If the margin $|1 - \langle x, a_i \rangle|$ is sufficiently large, then moving

---

∗Authors sorted in alphabetical order. Full version of this paper (containing additional theoretical results, additional experiments, and missing proofs) available at https://arxiv.org/abs/1901.02871.

from $x$ to a nearby point $y$ should not affect the sign of $1 - \langle x, a_i \rangle$, and thus not change the gradient. Therefore, if samples $a_1, \ldots, a_n$ are sufficiently spread out in the space, then a large fraction of them should incur large margins, and thus have the same gradients when $x$ changes by a small amount.

**Example 2.** In revenue management problems, $f_i(x)$ represents the marginal profit of the $i$-th customer under bid-price strategy $x \in \mathbb{R}^d_+$ over $d$ items. In many applications (see Section 2.2), $\nabla f_i(x)$ only depends on customer $i$'s preferences under $x$.

If the bid-price vector $x \in \mathbb{R}^d_+$ changes by a small amount to $y$, then for a large fraction of customers $i$, their most profitable items may not change, and thus $\nabla f_i(x) \approx \nabla f_i(y)$. (Indeed, imagine if one of the items is Xbox, and its price drops by 5%, perhaps 90% of the customers will not change their minds about buying or not. We shall demonstrate this using real-life data.)

## 1.1 Our Results

We assume in this paper that, given any point $x \in \mathbb{R}^d$ and index $i \in [n]$, one can efficiently evaluate a "lingering radius" $\delta(x, i)$. The radius satisfies the condition that for every point $y$ that is within distance $\delta(x, i)$ from $x$, the stochastic gradient $\nabla f_i(y)$ is equal to $\nabla f_i(x)$. We make two remarks:

- We use "equal to" for the purpose of proving theoretical results. In practice and in our experiments, it suffices to use approximate equality such as $\|\nabla f_i(x) - \nabla f_i(y)\| \leq 10^{-10}$.

- By "efficient" we mean $\delta(x, i)$ is computable in the same complexity as evaluating $\nabla f_i(x)$. This is reasonable because when $\nabla f_i(x)$ is an explicit function of $x$, it is usually easy to tell how sensitive it is to the input $x$. (We shall include such an example in our experiments.)

If we denote by $B(x, r)$ the set of indices $j$ satisfying $\delta(x, j) < r$, and if we travel to some point $y$ that is at most distance $r$ from $x$, then we only need to re-evaluate the (stochastic) gradients $\nabla f_j(y)$ for $j \in B(x, r)$. Intuitively, one should expect $|B(x, r)|$ to grow as a function of $r$, and this is indeed the case – for instance, for the revenue management problem (see Section 5).

**Theory.** To present the simplest theoretical result, we modify gradient descent (GD) to take into account the lingering of gradients. At a high level, we run GD, but during its execution, we maintain a decomposition of the indices $\Lambda_0 \cup \cdots \cup \Lambda_t = \{1, 2, \ldots, n\}$ where $t$ is logarithmic in $n$. Now, whenever we need $\nabla f_i(x_k)$ for some $i \in \Lambda_p$, we approximate it by $\nabla f_i(x_{k'})$ for a point $k'$ that was visited at most $2^p$ steps ago. Our algorithm makes sure that such $\nabla f_i(x_{k'})$ is available in memory.

We prove that the performance of our algorithm depends on how $|B(x, r)|$ grows in $r$. Formally, let $T$ be the total number of stochastic gradient computations divided by $n$, and suppose $|B(x, r)| \leq O(r)$, i.e., it linearly scales in the radius $r$. Then, our algorithm finds a point $x$ with $f(x) - f(x^*) \leq 2^{-\Omega(T^{1/3})}$. In contrast, traditional GD satisfies $f(x) - f(x^*) \leq O(T^{-1})$.

In the full version of this paper, we also study the case when $|B(x, r)| \leq O(r^\beta)$ for an arbitrary constant $\beta \in (0, 1]$.

**Practice.** We also design an algorithm that practically maximizes the use of gradient lingering. We take the SVRG method [19, 36] as the prototype because it is widely applied in large-scale settings. Recall that SVRG uses gradient estimator $\nabla f(\widetilde{x}) - \nabla f_i(\widetilde{x}) + \nabla f_i(x_k)$ to estimate the full gradient $\nabla f(x_k)$, where $\widetilde{x}$ is the so-called snapshot point (which was visited at most $n$ steps ago) and $i$ is a random index. At a high level, we modify SVRG so that the index $i$ is only generated from those whose stochastic gradients need to be recomputed, and ignore those such that $\nabla f_i(x_k) = \nabla f_i(\widetilde{x})$. This can further reduce the variance of the gradient estimator, and improve the running time.

## 1.2 Related Work

**Variance Reduction.** The SVRG method was independently proposed by Johnson and Zhang [19], Zhang et al. [36], and belong to the class of stochastic methods using the so-called variance-reduction technique [4, 8, 19, 23, 27–30, 35, 36]. The common idea behind these methods is to use some full gradient of the past to approximate future, but they do not distinguish which $\nabla f_i(x)$ can "linger longer in time" among all indices $i \in [n]$ for different $x$.

Arguably the two most widely applied variance-reduction methods are SVRG and SAGA [8]. They have complementary performance depending on the internal structural of the dataset [5], so we compare to both in our experiments.

**Reuse Gradients.** Some researchers have exploited the internal structure of the dataset to speed up first-order methods. That is, they use $\nabla f_i(x)$ to approximate $\nabla f_j(x)$ when the two data samples $i$ and $j$ are sufficiently close. This is orthogonal to our setting because we use $\nabla f_i(x)$ to approximate $\nabla f_i(y)$ when $x$ and $y$ are sufficiently close. In the extreme case when all the data samples are identical, they have $\nabla f_i(x) = \nabla f_j(x)$ for every $i, j$ and thus stochastic gradient methods converge as fast as full gradient ones. For this problem, Hofmann et al. [16] introduced a variant of SAGA, Allen-Zhu et al. [5] introduced a variant of SVRG and a variant of accelerated coordinate descent.

Other authors study how to reduce gradient computations at the snapshot points of SVRG [15, 20]. This is also orthogonal to the idea of this paper, and can be added to our algorithms for even better performance (see Section 5).

## 2  Notions and Problem Formulation

We denote by $\|\cdot\|$ the Euclidean norm, and $\|\cdot\|_\infty$ the infinity norm. Recall the notion of Lipschitz smoothness (it has other equivalent definitions, see textbook [25]).

**Definition 2.1.** *A function $f\colon \mathbb{R}^d \to \mathbb{R}$ is L-Lipschitz smooth (or L-smooth for short) if*
$$\forall x, y \in \mathbb{R}^d\colon \|\nabla f(x) - \nabla f(y)\| \le L\|x - y\| \ .$$

We also introduce the notion of "lowbit sequence" for a positive integer.

**Definition 2.2.** *For positive integer $k$, let $\mathsf{lowbit}(k) \overset{\text{def}}{=} 2^i$ where $i \ge 0$ is the maximum integer such that $k$ is integral multiple of $2^i$. For instance, $\mathsf{lowbit}(34) = 2$, $\mathsf{lowbit}(12) = 4$, and $\mathsf{lowbit}(8) = 8$.*

*Given positive integer $k$, let the $\mathsf{lowbit}$ sequence of $k$ be $(k_0, k_1, \dots, k_t)$ where*
$$0 = k_0 < k_1 < \cdots < k_t = k \quad \text{and} \quad k_{i-1} = k_i - \mathsf{lowbit}(k_i) \ .$$
*For instance, the lowbit sequence of $45$ is $(0, 32, 40, 44, 45)$.*

### 2.1  Our Model

We propose the following model to capture the lingering of gradients. For every $x \in \mathbb{R}^d$ and index $i \in [n]$, let $\delta(x, i) \ge 0$ be the *lingering radius* of $\nabla f_i(x)$, meaning that[2]
$$\forall y \in \mathbb{R}^d \text{ with } \|y - x\| \le \delta(x, i) \quad \text{it satisfies} \quad \nabla f_i(x) = \nabla f_i(y) \ .$$
In other words, as long as we travel within distance $\delta(x, i)$ from $x$, the gradient $\nabla f_i(x)$ can be reused to represent $\nabla f_i(y)$. Accordingly, for every $x \in \mathbb{R}^d$ and $r \ge 0$, we denote by $B(x, r)$ the set of indices $j$ satisfying $\delta(x, j) < r$. That is, $B(x, r) \overset{\text{def}}{=} \big\{ j \in [n] \,\big|\, \delta(x, j) < r \big\} \ .$

Our main assumption of this paper is that

**Assumption 1.** *Each $\delta(x, i)$ can be computed in the same time complexity as $\nabla f_i(x)$.*

Under Assumption 1, if at some point $x$ we have already computed $\nabla f_i(x)$ for all $i \in [n]$, then we can compute $\delta(x, i)$ as well for every $i \in [n]$, and sort the indices $i \in [n]$ in the increasing order of $\delta(x, i)$. In the future, if we arrive at any point $y$, we can calculate $r = \|x - y\|$ and use
$$\nabla' = \tfrac{1}{n}\Big( \textstyle\sum_{i \notin B(x,r)} \nabla f_i(x) + \sum_{i \in B(x,r)} \nabla f_i(y) \Big)$$
to represent $\nabla f(y)$. We stress that the time to compute $\nabla'$ is only proportional to $|B(x, r)|$.

We denote by $T_{\mathsf{time}}$ the gradient complexity, which equals how many times $\nabla f_i(x)$ and $\delta(x, i)$ are calculated, divided by $n$. In computing $\nabla'$ above, the gradient complexity is $|B(x, r)|/n$. If we always set $\delta(x, i) = 0$ then $|B(x, r)| = n$ and the gradient complexity for computing $\nabla'$ remains 1. However, if the underlying Problem (1.1) is nice enough so that $|B(x, r)|$ becomes an increasing function of $r$ (see Figure 2), then we can hope to design faster algorithms.

## 2.2 Revenue Management Problem

As a motivating example, consider a canonical revenue management problem of selling $d$ resources to $n$ customers. Let $b_j \geq 0$ be the capacity of resource $j \in [d]$; let $p_{i,j} \in [0,1]$ be the probability that customer $i \in [n]$ will purchase a unit of resource $j$ if offered resource $j$; and let $r_j$ be the revenue for each unit of resource $j$. We want to offer each customer one and only one candidate resource, and let $y_{i,j}$ be the probability we offer customer $i$ resource $j$. The following is an LP relaxation for this problem:

$$\max_{y \geq 0} \left\{ \sum_{i \in [n], j \in [d]} r_j p_{i,j} y_{i,j} \,\middle|\, \forall j \in [d], \sum_{i \in [n]} p_{i,j} y_{i,j} \leq b_j \bigwedge \forall i \in [n], \sum_{j \in [d]} y_{i,j} = 1 \right\} \quad (2.1)$$

This LP (2.1) and its variants have repeatedly found many applications, including adwords/ad allocation problems [3, 10, 11, 14, 17, 24, 34, 37], and revenue management for airline and service industries [7, 13, 18, 26, 31, 33]. Some authors also study the online version of solving such LPs [1, 2, 9, 12].

A standard way to reduce (2.1) to convex optimization is by regularization (cf. [37]). Let us subtract the maximization objective by $R(x) \overset{\text{def}}{=} \mu \sum_{i \in [n]} \overline{p_i} \sum_{j \in [d]} y_{i,j} \log y_{i,j}$, where $\overline{p_i} \overset{\text{def}}{=} \max_{i \in [n]} p_{i,j}$ and $\mu > 0$ is some small regularization weight. Then, after transforming to the dual, we have

$$\min_{x \geq 0} \left\{ \mu \sum_{i=1}^{n} \overline{p_i} \cdot \log Z_i + \sum_{j=1}^{d} x_j b_j \right\} \quad \text{where} \quad Z_i = \sum_{j=1}^{d} \exp\left( \frac{(r_j - x_j) p_{i,j}}{\overline{p_i} \mu} \right) . \quad (2.2)$$

Any solution $x$ (usually known as the *bid price* in operations management [32]) to (2.2) naturally gives back a solution $y$ for the primal (2.1), by setting

$$y_{i,j} = \exp\left( \frac{(r_j - x_j) p_{i,j}}{\overline{p_i} \mu} \right) / Z_i. \quad (2.3)$$

If we let $f_i(x) \overset{\text{def}}{=} \mu n \overline{p_i} \cdot \log Z_i + \langle x, b \rangle$, then (2.2) reduces to Problem (1.1). We conduct empirical studies on this revenue management problem in Section 5.

## 3 Our Modification to Gradient Descent

In this section, we consider a convex function $f(x) = \frac{1}{n} \sum_{i=1}^{n} f_i(x)$ that is $L$-smooth. Recall from textbooks (e.g., [25]) that if gradient descent (GD) is applied for $T$ iterations, starting at $x_0 \in \mathbb{R}^d$, then we can arrive at a point $x$ with $f(x) - f(x^*) \leq O\left( \frac{\|x_0 - x^*\|^2}{T} \right)$. This is the $\frac{1}{T}$ convergence rate.

To improve on this theoretical rate, we make the following assumption on $B(x, r)$:

**Assumption 2.** *There exists $\alpha \in [0, 1], C > 0$ such that,*

$$\forall x \in \mathbb{R}^d, r \geq 0: \quad \frac{|B(x, r)|}{n} \leq \psi(r) \overset{\text{def}}{=} \max\{\alpha, r/C\} .$$

It says that $|B(n, r)|$ is a growing function in $r$, and the growth rate is $\propto r$. (In the full version we investigate the more general case where the growth rate is $\propto r^\beta$ for arbitrary $\beta \in (0, 1]$.) We also allow an additive term $\alpha$ to cover the case that an $\alpha$ fraction of the stochastic gradients always need to be recalculated, regardless of the distance. We illustrate the meaningfulness of Assumption 2 in Figure 2.

Our result of this section can be summarized as follows. Hiding $\|x_0 - x^*\|, L, C$ in the big-$O$ notion, and letting $T_{\text{time}}$ be the gradient complexity, we can modify GD so that it finds a point $x$ with

$$f(x) - f(x^*) \leq O\left( \frac{\alpha}{T_{\text{time}}} + 2^{-\Omega(T_{\text{time}})^{1/3}} \right) .$$

We emphasize that our modified algorithm does not need to know $\alpha$ or $C$.

### 3.1 Algorithm Description

In classical gradient descent (GD), starting from $x_0 \in \mathbb{R}^d$, one iteratively updates $x_{k+1} \leftarrow x_k - \frac{1}{L} \nabla f(x_k)$. We propose $\text{GD}^{\text{lin}}$ (see Algorithm 1) which, at a high level, differs from GD in two ways:

**Algorithm 1** $\texttt{GD}^{\texttt{lin}}(f, x^{(0)}, S, C, D)$

---

**Input:** $f(x) = \frac{1}{n}\sum_{i=1}^{n} f_i(x)$ convex and $L$-smooth, starting vector $x^{(0)} \in \mathbb{R}^d$, number of epochs $S \geq 1$, parameters $C, D > 0$.
**Output:** vector $x \in \mathbb{R}^d$.
1: **for** $s \leftarrow 1$ **to** $S$ **do**
2:     $x_0 \leftarrow x^{(s-1)}$; $m \leftarrow \lceil (1 + \frac{C^2}{16D^2})^s \rceil$; and $\xi \leftarrow \frac{C}{m}$.
3:     $\mathbf{g} \leftarrow \vec{0}$ and $\mathbf{g}_i \leftarrow \vec{0}$ for each $i \in [n]$.
4:     **for** $k \leftarrow 0$ **to** $m - 1$ **do**
5:        Calculate $\Lambda_k \subseteq [n]$ from $x_0, \dots, x_k$ according to Definition 3.1.
6:        **for** $i \in \Lambda_k$ **do**
7:          $\mathbf{g} \leftarrow \mathbf{g} + \frac{\nabla f_i(x_k) - \mathbf{g}_i}{n}$ and $\mathbf{g}_i \leftarrow \nabla f_i(x_k)$.
8:        $x_{k+1} \leftarrow x_k - \min\{\frac{\xi}{\|\mathbf{g}\|}, \frac{1}{L}\}\mathbf{g}$              $\diamond$ *it satisfies* $\mathbf{g} = \nabla f(x_k)$
9:     $x^{(s)} \leftarrow x_m$;
10: **return** $x = x^{(S)}$.

---

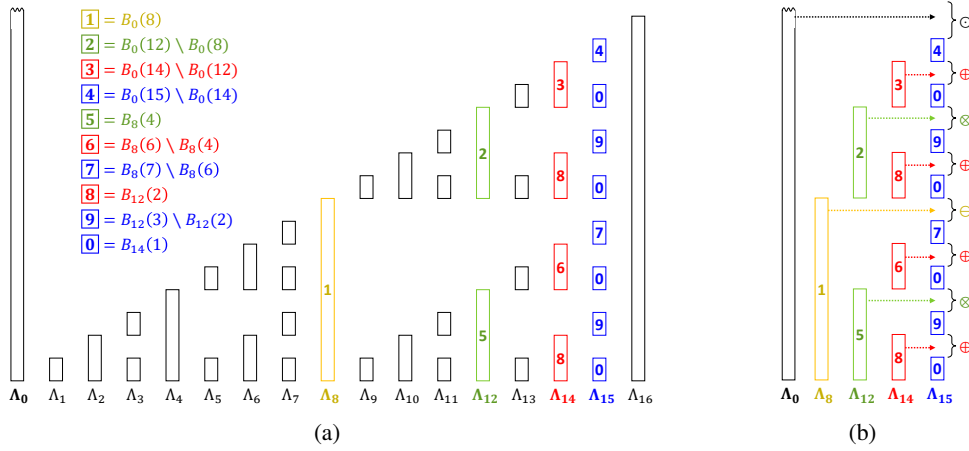

Figure 1: Illustration of index sets $\Lambda_k$

- It performs a truncated gradient descent with travel distance $\|x_k - x_{k+1}\| \leq \xi$ per step.
- It speeds up the process of calculating $\nabla f(x_k)$ by using the lingering of past gradients.

Formally, $\texttt{GD}^{\texttt{lin}}$ consists of $S$ epochs $s = 1, 2, \dots, S$ of growing length $m = \lceil (1 + \frac{C^2}{16D^2})^s \rceil$. In each epoch, it starts with $x_0 \in \mathbb{R}^d$ and performs $m$ truncated gradient descent steps

$$x_{k+1} \leftarrow x_k - \min\left\{ \frac{\xi}{\|\nabla f(x_k)\|}, \frac{1}{L} \right\} \cdot \nabla f(x_k) \ .$$

We choose $\xi = C/m$ to ensure that the worst-case travel distance $\|x_m - x_0\|$ is at most $m\xi = C$.

In each iteration $k = 0, 1, \dots, m-1$ of this epoch $s$, in order to calculate $\nabla f(x_k)$, $\texttt{GD}^{\texttt{lin}}$ constructs index sets $\Lambda_0, \Lambda_1, \dots, \Lambda_{m-1} \subseteq [n]$ and recalculates only $\nabla f_i(x_k)$ for those $i \in \Lambda_k$. We formally introduce index sets below, and illustrate them in Figure 1.

**Definition 3.1.** *Given* $x_0, x_1, \dots, x_{m-1} \in \mathbb{R}^d$, *we define index subsets* $\Lambda_0, \dots \Lambda_{m-1} \subseteq [n]$ *as follows. Let* $\Lambda_0 = [n]$. *For each* $k \in \{1, 2, \dots, m-1\}$, *if* $(k_0, \dots, k_t)$ *is* $k$'s *lowbit sequence from Definition 2.2, then (recalling* $k = k_t$)

$$\Lambda_k \overset{\text{def}}{=} \bigcup_{i=0}^{t-1} \left( B_{k_i}(k - k_i) \setminus B_{k_i}(k_{t-1} - k_i) \right) \quad \text{where} \quad B_k(r) \overset{\text{def}}{=} \Lambda_k \cap B(x_k, r \cdot \xi) \ .$$

## 3.2 Intuitions & Properties of Index Sets

We show in this paper that our construction of index sets satisfy the following three properties.

**Lemma 3.2.** *The construction of* $\Lambda_0, \dots, \Lambda_{m-1}$ *ensures that* $\mathbf{g} = \nabla f(x_k)$ *in each iteration* $k$.

**Claim 3.3.** *The gradient complexity to construct $\Lambda_0, \ldots, \Lambda_{m-1}$ is $O\left(\frac{1}{n}\sum_{k=0}^{m-1}|\Lambda_k|\right)$ under Assumption 1. The space complexity is $O(n\log n)$.*

**Lemma 3.4.** *Under Assumption 2, we have $\frac{1}{n}\sum_{k=0}^{m-1}|\Lambda_k| \leq O(\alpha m + \log^2 m)$.*

Claim 3.3 is easy to verify. Indeed, for each $\Lambda_\ell$ that is calculated, we can sort its indices $j \in \Lambda_\ell$ in the increasing order of $\delta(x_k, j)$.[3] Now, whenever we calculate $B_{k_i}(k-k_i) \setminus B_{k_i}(k_{t-1}-k_i)$, we have already sorted the indices in $\Lambda_{k_i}$, so can directly retrieve those $j$ with $\delta(x_{k_i}, j) \in (k_{t-1}-k_i, k-k_i]$.

As for the space complexity, in any iteration $k$, we only need to store $\lceil \log_2 k \rceil$ index sets $\Lambda_\ell$ for $\ell < k$. For instance, when calculating $\Lambda_{15}$ (see Figure 1(b)), we only need to use $\Lambda_0, \Lambda_8, \Lambda_{12}, \Lambda_{14}$; and from $k = 16$ onwards, we no longer need to store $\Lambda_1, \ldots, \Lambda_{15}$.

Lemma 3.2 is technically involved to prove (see full version), but we give a sketched proof by picture. Take $k = 15$ as an example. As illustrated by Figure 1(b), for every $j \in [n]$,

- If $j$ belongs to $\Lambda_{15}$ —i.e., boxes $4, 0, 9, 7$ of Figure 1—
  We have calculated $\nabla f_j(x_k)$ so are fine.
- If $j$ belongs to $\Lambda_{14} \setminus B_{14}(1)$ —i.e., $\oplus$ region of Figure 1(b)—
  We have $\nabla f_j(x_{15}) = \nabla f_j(x_{14})$ because $\|x_{15} - x_{14}\| \leq \xi$ and $j \notin B_{14}(1)$. Therefore, we can safely retrieve $\mathbf{g}_j = \nabla f_j(x_{14})$ to represent $\nabla f_j(x_{15})$.
- If $j$ belongs to $\Lambda_{12} \setminus B_{12}(3)$ —i.e., $\otimes$ region of Figure 1(b)—
  We have $\nabla f_j(x_{15}) = \nabla f_j(x_{12})$ for similar reason above. Also, the most recent update of $\mathbf{g}_j$ was at iteration 12, so we can safely retrieve $\mathbf{g}_j$ to represent $\nabla f_j(x_{15})$.
- And so on.

In sum, for all indices $j \in [n]$, we have $\mathbf{g}_j = \nabla f_j(x_k)$ so $\mathbf{g} = \frac{\mathbf{g}_1 + \cdots + \mathbf{g}_n}{n}$ equals $\nabla f(x_k)$.

Lemma 3.4 is also involved to prove (see full version), but again should be intuitive from the picture. The indices in boxes $1, 2, 3, 4$ of Figure 1 are disjoint, and belong to $B(x_0, 15\xi)$, totaling at most $|B(x_0, 15\xi)| \leq n\psi(15\xi)$. The indices in boxes $5, 6, 7$ of Figure 1 are also disjoint, and belong to $B(x_8, 7\xi)$, totaling at most $|B(x_8, 7\xi)| \leq n\psi(7\xi)$. If we sum up the cardinality of these boxes by carefully grouping them in this manner, then we can prove Lemma 3.4 using Assumption 2.

## 3.3 Convergence Theorem

So far, Lemma 3.4 shows we can reduce the gradient complexity from $O(m)$ to $\widetilde{O}(1)$ for every $m$ steps of gradient descent. Therefore, we wish to set $m$ as large as possible, or equivalently $\xi = C/m$ as small as possible. Unfortunately, when $\xi$ is too small, it will impact the performance of truncated gradient descent (see full version). This motivates us to start with small value of $m$ and increase it epoch by epoch. Indeed, as the number of epoch grows, $f(x_0)$ becomes closer to the minimum $f(x^*)$, and thus we can choose smaller values of $\xi$.

Formally, we have

**Theorem 3.5.** *Given any $x^{(0)} \in \mathbb{R}^d$ and $D > 0$ that is an upper bound on $\|x^{(0)} - x^*\|$. Suppose Assumption 1 and 2 are satisfied with parameters $C \in (0, D], \alpha \in [0, 1]$. Then, denoting by $m_s = \lceil (1 + \frac{C^2}{16D^2})^s \rceil$, we have that $\mathtt{GD}^{\mathtt{lin}}(f, x_0, S, C, D)$ outputs a point $x \in \mathbb{R}^d$ satisfying $f(x) - f(x^*) \leq \frac{4LD^2}{m_S}$ with gradient complexity $T_{\mathsf{time}} = O\left(\sum_{s=1}^{S}\alpha m_s + \log^2 m_s\right)$.*

As simple corollaries, we have

**Theorem 3.6.** *In the setting of Theorem 3.5, given any $T \geq 1$, one can choose $S$ so that $\mathtt{GD}^{\mathtt{lin}}$ finds a point $x$ in gradient complexity $T_{\mathsf{time}} = O(T)$ s.t.*

$$f(x) - f(x^*) \leq O\left(\frac{LD^4}{C^2} \cdot \frac{\alpha}{T}\right) + \frac{LD^2}{2^{\Omega(C^2 T/D^2)^{1/3}}} \ .$$

We remark here if $\psi(r) = 1$ (so there is no lingering effect for gradients), we can choose $C = D$ and in this case $\mathtt{GD}^{\mathtt{lin}}$ gives back the convergence $f(x) - f(x^*) \leq O\left(\frac{LD^2}{T}\right)$ of GD.

## 4  Our Modification to SVRG

In this section, we use Assumption 1 to improve the running time of SVRG [19, 36], one of the most widely applied stochastic gradient methods in large-scale settings. The purpose of this section is to construct an algorithm that works well *in practice*: to (1) work for any possible lingering radii $\delta(x,i)$, (2) be identical to SVRG if $\delta(x,i) \equiv 0$, and (3) be faster than SVRG when $\delta(x,i)$ is large.

Recall how the SVRG method works. Each *epoch* of SVRG consists of $m$ iterations ($m = 2n$ in practice). Each epoch starts with a point $x_0$ (known as the *snapshot*) where the full gradient $\nabla f(x_0)$ is computed exactly. In each iteration $k = 0, 1, \ldots, m-1$ of this epoch, SVRG updates $x_{k+1} \leftarrow x_k - \eta \mathbf{g}$ where $\eta > 0$ is the learning rate and $\mathbf{g}$ is the gradient estimator $\mathbf{g} = \nabla f(x_0) + \nabla f_i(x_k) - \nabla f_i(x_0)$ for some $i$ randomly drawn from $[n]$. Note that it satisfies $\mathbb{E}_i[\mathbf{g}] = \nabla f(x_k)$ so $\mathbf{g}$ is an unbiased estimator of the gradient. In the next epoch, SVRG starts with $x_m$ of the previous epoch.[4] We denote by $x^{(s)}$ the value of $x_0$ at the beginning of epoch $s = 0, 1, 2, \ldots, S-1$.

**Our Algorithm.**   Our algorithm $\mathtt{SVRG}^{\mathtt{lin}}$ (pseudocode in full version) maintains *disjoint* subsets $H_s \subseteq [n]$, where each $H_s$ includes the set of the indices $i$ whose gradients $\nabla f_i(x^{(s)})$ from epoch $s$ can still be safely reused at present.

At the starting point $x_0$ of an epoch $s$, we let $H_s = [n] \setminus (H_0 \cup \cdots \cup H_{s-1})$ and re-calculate gradients $\nabla f_i(x_0)$ only for $i \in H_s$; the remaining ones can be loaded from the memory. This computes the full gradient $\nabla f(x_0)$. Then, we denote by $m = 2|H_s|$ and perform only $m$ iterations within epoch $s$. We next discuss how to perform update $x_k \to x_{k+1}$ and maintain $\{H_s\}_s$ during each iteration.

- In each iteration $k$ of this epoch, we claim that $\nabla f_i(x_k) = \nabla f_i(x_0)$ for every $i \in H_0 \cup \cdots \cup H_s$.[5] Thus, we can uniformly sample $i$ from $[n] \setminus (H_0 \cup \cdots \cup H_s)$, and construct an unbiased estimator

$$\mathbf{g} \leftarrow \nabla f(x_0) + \left( 1 - \frac{\sum_{s'=0}^{s} |H_{s'}|}{n} \right) [\nabla f_i(x_k) - \nabla f_i(x_0)]$$

  of the true gradient $\nabla f(x_k)$. Then, we update $x_{k+1} \leftarrow x_k - \eta \mathbf{g}$ the same way as SVRG. We emphasize that the above choice of $\mathbf{g}$ reduces its variance (because there are fewer random choices), and it is known that reducing variance leads to faster running time [19].
- As for how to maintain $\{H_s\}_s$, in each iteration $k$ after $x_{k+1}$ is computed, for every $s' \leq s$, we wish to remove those indices $i \in H_{s'}$ such that the current position $x$ lies outside of the lingering radius of $i$, i.e., $\delta(x^{(s)}, i) < \|x - x^{(s)}\|$. To efficiently implement this, we need to make sure that whenever $H_{s'}$ is constructed (at the beginning of epoch $s'$), the algorithm sort all the indices $i \in H_{s'}$ by the increasing order of $\delta(x^{(s')}, i)$. We include implementation details in full version.

## 5  Preliminary Empirical Evaluation

In this section, we construct a revenue maximization LP (2.1) using the publicly accessible dataset of Yahoo! Front Page Today Module [6, 22]. We describe details of the experimental setup in full version. Based on this real-life dataset, we validate Assumption 2 and our motivation behind lingering gradients. We also test the performance of $\mathtt{SVRG}^{\mathtt{lin}}$ from Section 4 on optimizing this LP.

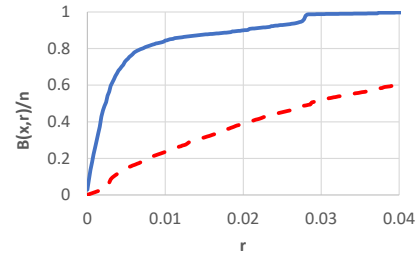

Figure 2: $|B(x, r)|/n$ as a function of $r$. We choose $\theta = 5$. Dashed curve is when $x = \vec{0}$, and solid curve is when $x$ is near the optimum.

**Illustration of Lingering Radius.**   We calculate lingering radii on the dual problem (2.2). Let $\theta > 0$ be a parameter large enough so that $e^{-\theta}$ can be viewed as zero. (For instance, $\theta = 20$ gives $e^{-20} \approx 2 \times 10^{-9}$.) Then, for each point $x \in \mathbb{R}_{\geq 0}$ and index $i \in [n]$, we let

$$\delta(x, i) = \min_{j \in [n], j \neq j^*} \frac{(r_{j^*} - x_{j^*})p_{i,j^*} - (r_j - x_j)p_{i,j} - \theta \overline{p_i} \mu}{p_{i,j^*} + p_{i,j}} \quad \text{where} \quad j^* = \arg\max_{j \in [n]} \left\{ (r_j - x_j)p_{i,j} \right\} .$$

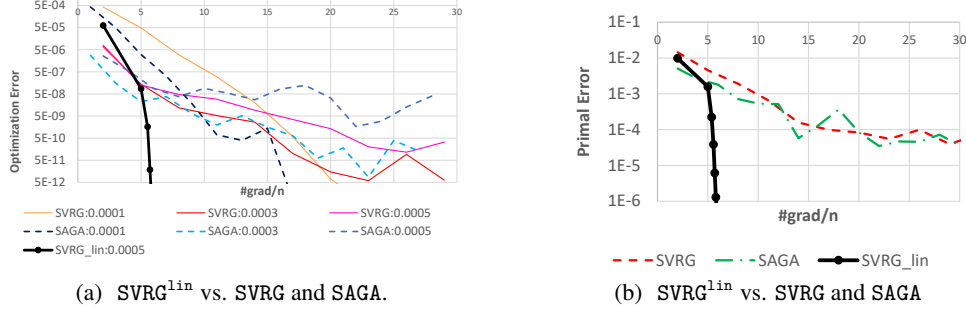

| (a) SVRG$^{\text{lin}}$ vs. SVRG and SAGA. | (b) SVRG$^{\text{lin}}$ vs. SVRG and SAGA |

Figure 3: Comparison of (a) dual objective optimality (for which different learning rates are presented) and (b) primal objective optimality (for which the learning rates are best tuned).

It is now a simple exercise to verify that, denoting by $\mathbf{e}_j$ the $j$-th basis unit vector, then[6]

$$\nabla f_i(y) \approx (b_1, \ldots, b_d) + n p_{i,j^*} \mathbf{e}_{j^*} \quad \text{for every} \quad \|y - x\|_\infty \le \delta(x, i) \ .$$

In Figure 2, we plot $|B(x, r)| = \left| \left\{ j \in [n] \,\middle|\, \delta(x, j) < r \right\} \right|$ as an increasing function of $r$. We see that for practical data, $|B(x, r)|/n$ is indeed bounded above by some increasing function $\psi(\cdot)$. We provide more justification on why this happens in the full paper.

*Remark* 5.1. This $\delta(x, i)$ differs from our definition in Section 2 in two ways. First, it ensures $\nabla f_i(y) \approx \nabla f_i(x)$ as opposed to exact equality; for practical purposes this is no big issue, and we choose $\theta = 5$ in our experiments. Second, $\|y - x\|_\infty \le \delta(x, i)$ gives a bigger "safe region" than $\|y - x\| \le \delta(x, i)$; thus, when implementing SVRG$^{\text{lin}}$, we adopt $\| \cdot \|_\infty$ as the norm of choice.

**Numerical Experiments.** We consider solving the dual problem (2.2). In Figure 3(a), we plot the optimization error of (2.2) as a function $\#\text{grad}/n$, the number of stochastic gradient computations divided by $n$, also known as $\#$*passes of dataset*.

Figure 3(a) compares our SVRG$^{\text{lin}}$ to SVRG and SAGA (each for 3 best tuned step lengths).[7] We can see SVRG$^{\text{lin}}$ is close to SVRG or SAGA during the first 5-7 passes of the data. This is because initially, $x$ moves fast and cannot usually stay in the lingering radii for most indices $i$. After that period, SVRG$^{\text{lin}}$ requires a dramatically smaller number of gradient computations, as $x$ moves slower and slower, becoming more easily to stay in the lingering radii. It is interesting to note that SVRG$^{\text{lin}}$ does not significantly improve the optimization error as a function of number of epochs; the improvement primarily lies in improving the number of gradient computations per epoch. The comparison is

**More Plots.** In Figure 3(b), we also compare the primal objective value for the LP (2.1). (We explain how to get feasible primal solutions from the dual in the full version.) It is perhaps worth noting that we have chosen $\mu = 10^{-5}$ as the regularization error, and the primal objective error indeed reaches to $10^{-6}$ which is roughly $\mu$. In the full version, we also compare the running time of the algorithms. Those plots are almost identical to Figure 3(b).

## 6 Conclusion

In this paper, we study convex problems where the stochastic gradients $\nabla f_i(x)$ can be reused when we move away from $x$. In our theoretical result, we model the number of stochastic gradients that can be changed (and thus cannot be reused) as a function of how much distance we travel away from $x$, and show faster convergence for gradient descent (in terms of the number of gradient computations). On the empirical side, we show how to modify the SVRG method to use reuse stochastic gradients efficiently. Figure 3(a) and Figure 3(b) summarize our findings on a hypothetic experiment.

## Acknowledgements

We would like to thank Greg Yang, Ilya Razenshteyn and Sébastien Bubeck for discussing the motivation of this problem.

## Footnotes

[2]Recall that, in practice, one should replace the exact equality with, for instance, $\|\nabla f_i(x) - \nabla f_i(y)\| \le 10^{-10}$. To present the simplest statements, we do not introduce such an extra parameter.

[3] Calculating those lingering radii $\delta(x_k, j)$ require gradient complexity $|\Lambda_\ell|$ according to Assumption 1, and the time for sorting is negligible.

[4]Some authors use the average of $x_1, \ldots, x_m$ to start the next epoch, but we choose this simpler version.

[5]This is because for every $i \in H_s$, by definition of $H_s$ we have $\nabla f_i(x_k) = \nabla f_i(x^{(s)}) = \nabla f_i(x_0)$; for every $i \in H_{s'}$ where $s' < s$, we know $\nabla f_i(x_k) = \nabla f_i(x^{(s')})$ but we also have $\nabla f_i(x_0) = \nabla f_i(x^{(s')})$ (because otherwise $i$ would have been removed from $H_{s'}$).

[6]For any other coordinate $j \ne j^*$, it satisfies $\frac{e^{(r_j - y_j) p_{i,j} / (\overline{p_i} \mu)}}{e^{(r_{j^*} - y_{j^*}) p_{i,j^*} / (\overline{p_i} \mu)}} \le e^{-\theta}$ and hence is negligible.

[7]Each epoch of SVRG consists of a full gradient computation and $2n$ iterations, totaling $3n$ computations of (new) stochastic gradients. (We do not count the computation of $\nabla f_i(0)$ at $x = 0$.)

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
