[Reviews · NeurIPS 2018]

Reviewer 1



The main idea of the paper is to use continuity properties of the gradient of a function to "reuse" gradients i.e. for nearby points, one may reuse gradients that have been already computed. The paper develops a conceptual and algorithmic framework to exploit this idea. Pros: 1) The idea of lingering gradients is an interesting concept 2) nice algorithm/data structure to exploit lingering gradients 3) extension of 2) to SVRG and the discussion on implementation details nicely done. Promising numerical results. However, some major issues: Larger problems: With the assumptions they make this is a solid paper in my opinion. However, I'm not so about the validity and relevance of these assumptions. In abstract they claim that the algorithm has very good performance under certain conditions (alpha = 0 in their analysis if I understod correctly). To me it seems like that the only cases where this condition applies is when all component functions are constants. When motivating assumption 2 they show a plots of |B(x,r)| for two different x and say that the plot shows that indicates that assumption 2 is true. The hard part about this assumption is that |B(x,r)| should be uniformly bounded for all x. On the contrary it seems to me that the plot indicates that as we approach the optimum to bound |B(x,r)| with r^(b)/C either b or C has to decrease. It should be said that the method performs very well on this example so it should be possible to verify the assumptions in this case. They have two examples in the introduction where they discuss when this method might be relevant. It seems to me that the first example (svm) is of a lot more interest to the ml community than a resource allocation problem so the choice of numerical experiment is a bit strange. Some minor things: In theorem 3.5 I believe that D should be an upper bound of ||x0 - x*|| and not ||x0 - x*||^2. Figure 1 uses the '0' label for two different things. Should probably be a 10 instead. Some inconsistency in how they present performance with regards to step lengths in figure 3a (seems weird to only present different step lengths for some of the algorithms). In the end, the value of this paper to me really comes down to whether assumption 1 and 2 are relevant in ml applications. After reading the paper I'm not convinced that they are.

Reviewer 2



Summary: -------- The paper studies very lazy updates of stochastic gradients where one uses previous computed gradients instead of computing new ones. They justify such approach by introducing the notion of "lingering gradients", that they define properly. A theoretical approach is provided, assuming this new property, the authors propose a careful storage of previous gradients to exploit this property. Then experiments are made on a simpler method based on SVRG that, while loosing the theoretical properties previously proven, show strong improvements of their approach compared to classical ones. Quality and clarity: -------------------- The paper is very well written. Motivation is clear and outline well carried on. The proofs are complicated but readable and correct (though I put below details that I''d like to be clarified). Such quality of paper offers the possibility to open discussions in conference. Originality and significance: ----------------------------- The paper gives theoretical tools to tackle a heuristic that was interesting in practice and where they show real improvements. The storage of gradients using appropriate structure is original to me and may be worth disseminating. Nevertheless, the original definition of lingering gradients seems quite unrealistic. The natural application of this framework is non-smooth optimization, where in some regions, the gradient indeed doesn't change. However characterizing how these regions smoothly change seems impossible. It seems to me that this lingering gradient is an approximation for very smooth functions (Lipschitz constant very small). -> Could the authors relate the approximate lingering property (up to a given epsilon) to the simple smoothness of the gradient in the experiments they do ? In other words can these Lipschitz constant be computed to understand better the approximation ? Note that Holder smoothness of the function would also be interesting to relate the the beta exponent of their definition. It seems to me that in such cases it would be interesting to relate the proposed approach to approximate gradient methods that exploit the Holder smoothness see e.g. [Y. Nesterov 2013, Universal Gradient Methods for Convex Optimization Problems] but in a stochastic case. Conclusion: ----------- Though I think the definition of lingering gradients is for the moment unrealistic, the proposed approach is an interesting step towards heuristics that use previous gradients in a lazy way. Overall the quality of the paper enables to open discussions in the conference that could be fruitful. Since the experiments part is particularly convincing, having the code available would definitely improve the paper (and so my overall appreciation). Proof check: ------------ l 427: Precise that you used B_{k_i}(k_{t-1}-k_i) \subset B_{k_i}(k_t-k_i) because k_t > k_{t-1} l 461: Could you precise the terms hidden in the O ? l 490, Bounds on T_{time}: Detailing the inequalities would ease the reading. In particular I don't get the last inequality: form the assumptions we have log(m_S) > log(T) Details: -------- l 117: the LP (2.1) is convex, the proposed formulation is therefore not a reduction to convex optimization but another smoothed formulation of the problem. l 490: Shouldn't it be m_S = \Omega(min(...,...)) without the \Theta ? l 490: In the case beta \in (0,1): erase additional parenthesis in the lower bound on m_S, correct m_s in m_S in the bound of T_{time}

Reviewer 3



The authors propose to modify stochastic gradient methods such as SVRG to reuse computed stochastic gradients. They made two assumptions: A1. When perturbing parameters with radius r, the fraction of stochastic gradients need to recompute grows sublinearly w.r.t. r. A2. Estimating a safe radius r w.r.t. some sample in A1 is as hard as computing its stochastic gradient. With these assumptions, the authors showed theoretically that gradient descent can be modified to achieve better gradient complexity (i.e. w.r.t. number of gradient computation). These assumptions are then verified empirically on a revenue maximization problem on real-world dataset. For experiments, the authors showed that SVRG as well as its SCSG variant can be modified to achieve better gradient complexity. Strengths: The paper is well written. The theoretical analysis is sound and the idea is novel to me. If the assumptions hold, this approach can be generalized to other problems. Weaknesses: 1. Generalizability. In general, I think the authors need to show how this approach can work on more problems. For example, it looks to me that for most deep net problem A2 is not true. Also, some empirical verification of assumption A1 alone on other problems would be useful to convince me why this approach can generalize. 2. Stability evaluation/analysis is missing. How sensitive is the performance to the lingering radius (i.e. theta or equivalently delta(x, i))? Could the authors give some theoretical analysis or some empirical evaluation? 3. Memory consumption. For many real-world applications, the stochastic gradient methods mentioned in this paper are not acceptable due to huge memory consumption. Could the authors explain how to generalize this approach to other methods, e.g. stochastic gradient descent with fixed batch size? I would expect the growing number of lingering stochastic gradients to be an issue. Some Typos: L532: difference --> different L540: report report --> report ------- Response to Author's feedback: 1. For A1, I agree that if we have explicit written f_i(x) then we can compute radius in a easy way. My original concern is when the function is too complicated that the radius do not have a easy close form, then can we at least empirically evaluate the radius. I guess if the authors want to focus outside DL then this might not be a big issue anymore. 2. I think my concern is no longer a issue if the function can be written explicitly. 3. Originally, I was imagining a deep net setting, where storing O(nd) is not acceptable. And I have concerns about the overhead of computing this on the fly. But I guess it's not a problem whenever SVRG is acceptable.